# Drivers and Distribution of Henipavirus-Induced Syncytia: What Do We Know?

**DOI:** 10.3390/v13091755

**Published:** 2021-09-02

**Authors:** Amandine Gamble, Yao Yu Yeo, Aubrey A. Butler, Hubert Tang, Celine E. Snedden, Christian T. Mason, David W. Buchholz, John Bingham, Hector C. Aguilar, James O. Lloyd-Smith

**Affiliations:** 1Department of Ecology & Evolutionary Biology, University of California Los Angeles, Los Angeles, CA 90095, USA; aabutler@g.ucla.edu (A.A.B.); htang662001@gmail.com (H.T.); csnedden@g.ucla.edu (C.E.S.); jlloydsmith@ucla.edu (J.O.L.-S.); 2Department of Microbiology & Immunology, Cornell University, Ithaca, NY 14850, USA; yy826@cornell.edu (Y.Y.Y.); dwb265@cornell.edu (D.W.B.); ha363@cornell.edu (H.C.A.); 3Department of Computational Medicine, University of California Los Angeles, Los Angeles, CA 90095, USA; ctmason@ucla.edu; 4CSIRO Australian Centre for Disease Preparedness, Geelong, VIC 3220, Australia; John.Bingham@csiro.au

**Keywords:** cell–cell fusion, henipavirus, pathogenesis, paramyxovirus, syncytium, within-host dynamics

## Abstract

Syncytium formation, i.e., cell–cell fusion resulting in the formation of multinucleated cells, is a hallmark of infection by paramyxoviruses and other pathogenic viruses. This natural mechanism has historically been a diagnostic marker for paramyxovirus infection in vivo and is now widely used for the study of virus-induced membrane fusion in vitro. However, the role of syncytium formation in within-host dissemination and pathogenicity of viruses remains poorly understood. The diversity of henipaviruses and their wide host range and tissue tropism make them particularly appropriate models with which to characterize the drivers of syncytium formation and the implications for virus fitness and pathogenicity. Based on the henipavirus literature, we summarized current knowledge on the mechanisms driving syncytium formation, mostly acquired from in vitro studies, and on the in vivo distribution of syncytia. While these data suggest that syncytium formation widely occurs across henipaviruses, hosts, and tissues, we identified important data gaps that undermined our understanding of the role of syncytium formation in virus pathogenesis. Based on these observations, we propose solutions of varying complexity to fill these data gaps, from better practices in data archiving and publication for in vivo studies, to experimental approaches in vitro.

## 1. Introduction

The formation of syncytia, or multinucleated cells induced by cell–cell fusion (Figure 1a), is characteristic of many viral families, including emerging and endemic *Herpesviridae* [1,2], *Retroviridae* [3,4], *Coronaviridae* [5,6,7], and *Paramyxoviridae* [8,9] among others. While syncytium formation is associated with many animal viruses, it remains unclear to what extent syncytia contribute to infection pathogenesis in vivo (i.e., disease development, including viral replication and spread within a host, and associated pathogenicity) [3,4,10]. This question is of particular interest today due to the diversity of emerging viruses that present cell–cell fusion as a distinguishing feature in vitro.

Syncytium formation is one of several mechanisms allowing direct cell-to-cell transmission of viral material without requiring the assembly and budding of free virions [11]. Other mechanisms of direct cell-to-cell transmission used by viruses include pore formation across tight junctions, neuronal and immunological synapses, or viral-induced filopodia [11,12]. Understanding the relationship between direct cell-to-cell transmission and pathogenesis can yield significant insights on the dynamics of viral infections. For instance, direct cell-to-cell transmission may significantly contribute to virus population growth, as observed in vitro for human immunodeficiency virus [13]. Cell-to-cell transmission can also be associated with altered dissemination patterns, ultimately impacting disease manifestation. For instance, hyperfusogenic mutants (exhibiting enhanced cell–cell fusion) of herpesviruses [14,15,16]. Similarly, in murine retrovirus infections, brain hemorrhages have been attributed to the alteration of the blood–brain barrier via the virus-induced fusion of endothelial cells [17,18]. In measles (MeV) infections, persistent brain infections leading to fatal subacute sclerosing panencephalitis have been attributed to synaptic transmission [19]. Hence, while free virions are crucial for viral transmission between organisms and viral spread within hosts, direct cell-to-cell transmissions may contribute greatly to pathogenesis. Yet, there remains a lack of empirical knowledge on the contribution of direct cell-to-cell transmission on viral propagation and disease development. Understanding the contribution of syncytium formation to pathogenesis would also inform on whether in vitro cell–cell fusion assays, which are widely used to functionally describe viruses [20,21,22,23,24], are useful predictors of in vivo dynamics.

The potential benefit of syncytium formation (and other direct cell-to-cell transmission mechanisms) for viruses has been attributed to efficient local transmission through the exploitation of preexisting cellular interactions such as cellular adhesion [12]. Indeed, during direct cell-to-cell transmission, the virion assembly and budding steps can be skipped, potentially accelerating the dissemination of viral material [11]. Direct cell-to-cell transmission may also facilitate evasion from the humoral immune response [25]. However, syncytium formation may also come with costs, such as inducing host cell apoptosis, as observed in MeV-infected immune cells [26,27,28,29], which may ultimately be detrimental to viral replication. It is thus not obvious whether syncytium formation has a positive, negative, or neutral effect on virus replication and dissemination. Unfortunately, in vitro studies of cell–cell fusion and free-virion production have historically been conducted independently [12,30], with the former being the focus of membrane fusion mechanistic studies (e.g., [22,31,32,33]) and the latter used to measure viral replication (by quantifying viral material in cell culture supernatant; e.g., [34,35]). In vivo, there has been no attempt to systematically map the occurrence of syncytia across virus and/or host species. Consequently, the relationship between syncytium formation and pathogenesis remains poorly understood.

Among the many emerging infectious viruses exhibiting syncytium formation, the *Henipavirus* genus (belonging to the *Paramyxoviridae* family) is of particular concern due to the high fatality rates of Hendra (HeV) and Nipah (NiV) viruses in humans [36]. The recent discovery of 20 novel henipaviruses (HNVs) in wild bats and rodents—including Cedar virus (CedV), Kumasi virus (KuV), and Mojiang virus (MojV) [37,38,39]—and evidence of spillovers to human populations [40] have further elevated concerns regarding the diversity and the zoonotic, pathogenic, and pandemic potentials of HNVs. Emerging evidence suggests that HNVs exhibit varying levels of pathogenicity in different hosts. For example, while HeV and NiV are highly pathogenic in susceptible hosts including humans [36], CedV appears non-pathogenic in animal models [34,38]. However, comparative studies involving multiple host and virus species are necessary to uncover the importance of different factors, such as syncytium formation, in HNV pathogenicity.

Although reviews of histopathological findings following HNV infections are available [34,41,42,43,44,45], it is difficult to compare results from different studies as it is unclear which samples were actually screened for syncytia. For instance, syncytia were reported in cats in endothelial and respiratory epithelial cells following NiV infection and only in endothelial cells following HeV infection [41], but it is not clear whether respiratory epithelial cells were even screened for syncytium formation following HeV infections. This lack of systematic reporting limits our ability to understand the contribution of syncytium formation in infection pathogenesis, and whether this trait can be used to assess the zoonotic and pathogenic potentials of emerging viruses. In this context, the objectives of this review are (1) to summarize current knowledge regarding the mechanisms driving syncytium formation following HNV infections, mostly acquired from in vitro experiments, and (2) to identify patterns of syncytium distribution in vivo based on a meta-analysis of literature data. We highlight the observed patterns and their concordance or discordance between in vitro and in vivo studies, discuss the fundamental and practical implications, and identify future directions for the study of virus-induced syncytium formation.

## 2. Insights on Syncytium Formation from In Vitro Studies

### 2.1. The Molecular Prerequisites of Syncytium Formation

HNV-induced membrane fusion, either at the virus–cell or cell–cell interface, is driven by the interactions between the viral attachment glycoprotein (G), the fusion glycoprotein (F, a class I fusion protein) and the host receptors—membrane proteins of the ephrin family (Figure 1c). Studies of the cascade of interactions between G, F, and ephrin receptors are dominated by in vitro experiments using cells transfected with wild-type or mutant G and F and measuring cell–cell fusion as an output (Box 1; Figure 2) [22,31,32,33,47]. Briefly, the binding of G to a compatible ephrin first induces conformational changes in G that expose a sequestered F-triggering region on its stalk domain, which then triggers the unfolding of F [31,32,48]. The fusion peptide of F anchors onto the target cell membrane and adopts a pre-hairpin intermediate conformation. This is followed by the coalescence of F that generates a six-helix bundle conformation, a process that induces hemifusion, pore formation (membrane fusion), and pore expansion [47,49]. The molecular mechanisms driving the transition from hemifusion to fusion pore formation and expansion remain poorly understood, despite the critical importance of these steps for completing syncytium formation.

Membrane fusion is thus highly dependent on host receptor expression. HNVs bind different ephrins with high specificities. HeV and NiV bind ephrin B2 and B3 [20,21,50,51] (although NiV has a higher affinity for ephrin B3 than HeV [51]), CedV binds ephrin A2, A5, B1, B2, and mouse A1 [52,53], and KuV binds ephrin B2 [54]. While receptor usage differences across host species are presumably minimal because ephrins are highly conserved [55,56], a notable exception is the ability of CedV to bind mouse but not human ephrin A1 due to a single amino-acid difference [52], which highlights the specificity of HNV receptor recognition. The receptor(s) for MojV remain elusive, though ephrin B2 and B3, sialic acid, and CD150—corresponding to the three receptor families currently known to be used by paramyxoviruses—have been ruled out [57].

Prior to membrane fusion, the proteolytic cleavage of immature F proteins (F_0_) into mature subunits (F_1−2_) is necessary (Figure 1b) to expose the concealed fusion peptide in the F_1_ region. F_0_ is cleaved by the cathepsin family proteases found in host cell endosomes. Cathepsin usage can differ between cell lines (e.g., cathepsin B in canine kidney MDCK cells [58], cathepsin L in monkey kidney Vero cells [59,60]). Cleavage occurs as F_0_ undergoes endosomal recycling [49], a constitutive process, thus enabling constant presentation of mature F_1−2_ at the surface of infected cells. HNV-induced membrane fusion is thus a pH-independent process, allowing membrane fusion to occur at the cell surface in physiologic conditions, a prerequisite for cell–cell fusion in vivo.

Beyond the need for host ephrin receptors and cathepsin proteases, recent studies suggest that HNV-induced membrane fusion is modulated by several other factors ranging from membrane composition to intrinsic properties of G and F. For instance, membrane cholesterol levels correlate with the intensity of cell–cell fusion induced by NiV glycoproteins [33]. Interactions with the cytoskeleton also modulate cell–cell fusion, as highlighted by proteomics [61]. The study of HNV mutants also revealed that the early and late steps of the membrane fusion cascade are modulated by several regions of G head and stalk domains [31,32,48] and of F [62,63]. Furthermore, N- and O-glycans in most paramyxoviral glycoproteins have been shown to modulate protein expression, folding, and transport; for HNVs in particular, several N- and O-glycans in NiV and HeV G and F modulate virus–cell and cell–cell fusion independently of protein expression [64,65,66]. It is worth noting that, although membrane fusion levels induced by NiV and HeV are relatively similar [24], some N- and O- glycans of NiV and HeV G and F modulate syncytium formation differently [67,68,69]. In addition, while cell–cell fusion and virus–cell fusion mechanisms are presumably driven by similar underlying mechanisms, incongruities have been reported. For instance, depleting NiV matrix protein reduced viral–cell fusion whilst enhancing cell–cell fusion [70]. Collectively, these studies demonstrate the complexity of the membrane fusion process and highlight the need for comparative studies considering both virus–cell and cell–cell fusion.

Box 1Methodological considerations of the study of syncytium formation in vitro and in vivo.Inducing cell–cell fusion in vitro. Syncytia can be observed in cell cultures infected with live viruses. Another approach with which to obtain syncytia in vitro, referred to as the “fusion assay”, consists in transfecting non-infected cells with plasmids encoding viral glycoproteins (called “effector” cells), and putting them in contact with susceptible cells (“target” cells; Figure 2) [20]. Cell–cell fusion can occur between effector and target cells obtained from different cell lines (e.g., [20]). This approach is limited to cell lines that are susceptible to transfection, and is thus challenging (and rarely implemented) in many relevant cell types.Opportunities from controlled in vitro conditions. Selectively transfecting genes into cells allows a high degree of control and eliminates other processes that may influence cell–cell fusion such as viral replication and budding. In contrast, live virus infection into cell culture requires successful infection and completion of a full replicative cycle, which ultimately affect cell–cell fusion via their impact on viral protein expression. More generally, in vitro studies enable a high degree of experimental flexibility to elucidate minute details relevant to cell–cell fusion that are not attainable in in vivo systems, either by controlling or including the quantification of specific variables. Examples include the ability to artificially engraft glycoproteins with reporter tags to monitor intracellular trafficking [71] and assess cell-surface expression [20], or the ability to explore the effect of specific mutations on membrane fusion [66], among others.Biological constraints specific to in vivo conditions. Additional factors are at play in vivo. First, tissue localization and connectivity can greatly impact the infection probability of specific tissues, and ultimately determine syncytium formation independently of cell susceptibility. Second, cell–cell fusion requires close and stable contact between cells [13]. While this is generally not an issue in vitro, it can have varying consequences in vivo. Cell motility may represent an obstacle to syncytium formation, while physiologic cellular interactions (e.g., between endothelial or epithelial cells [72]) may favor syncytium formation. Finally, host immunity may be a critical modulator of cell–cell fusion. For instance, neutralizing antibodies produced as part of adaptive immunity are likely to impact cell–cell fusion by binding viral glycoproteins at the surfaces of infected cells. Detecting and quantifying syncytia in vitro and in vivo. Cell–cell fusion in vivo is usually observed using classical histological approaches, whereby tissue samples collected from animals are fixed, thinly-sliced, stained, and visualized under a microscope (Figure 3). Meanwhile, because cell cultures in vitro are monolayers, it is possible to directly visualize cell–cell fusion without fixation or staining (Figure 2). Reporter genes (e.g., *β*-galactosidase [73], *β*-lactamase [66], dual-split protein [74] assays) are increasingly commonly used to assist in the detection of cell–cell fusion, making in vitro assays potentially more sensitive for syncytium detection and enabling a more accurate quantification of cell–cell fusion. Comparing intensities of syncytium formation between in vitro and in vivo conditions is thus challenging because the methods used for syncytium quantification are different, but also because the definition of syncytium differs. Some in vivo studies mention syncytia with 2 nuclei (e.g., [75]), while many in vitro studies using manual counting of syncytia (in opposition to reporter genes) only consider cells with four or more nuclei (e.g., [24,69]) as rare small nuclei can be observed in cell cultures independently of the presence of viral proteins.

### 2.2. Occurrence of Henipavirus-Induced Syncytia across Viruses and Cell Lines

In vitro studies have identified virus and host factors, such as ephrin binding of G and cathepsin cleavage of F, that may account for differences in susceptibility to syncytium formation across hosts and tissues. Syncytium formation has been observed for all five known HNVs in a wide range of cell lines including cells from the kidney, lung, brain, and vascular and lymphatic systems of primates, hamsters, or swine (Table 1). The main factor driving susceptibility to cell–cell fusion in vitro appears to be the expression of functional receptors. For instance, HeV, NiV, CedV, or KuV-induced cell–cell fusion in refractory cells can be rescued upon the introduction of a functional ephrin receptor [20,50,52,53,54].

Although a wide range of cell lines are susceptible to HNV-induced cell–cell fusion and qualitative patterns (susceptible versus refractory to cell–cell fusion) seem conserved across HNVs, quantitative variations have been reported. For instance, CedV induces significantly less cell–cell fusion than NiV in human kidney HEK293T cells [48]. Similarly, KuV and MojV also appear to form fewer syncytia than NiV in HEK293T and hamster kidney BHK cells [23,57]. Quantitative variations in syncytium formation have also been reported within the Nipah clade in some conditions [76], but not all [77]. However, as those studies were conducted on live viruses, it is not possible to tease apart differences in cell–cell fusion phenotype versus differences in replication kinetics (resulting in different syncytium formation kinetics). In any case, the mechanisms underlying quantitative variations across virus isolates and cells remain unclear due to the many factors involved in cell–cell fusion, as described above.

Furthermore, for a given virus, variations in syncytium formation also exist across cell lines. For instance, NiV glycoproteins produce more fusion in Vero than HEK293T cells [20]. HeV glycoproteins produce more fusion in rabbit kidney RK13 cells than Vero cells [73]. Infections of bat kidney PaKiT03 cells and HEK293T cells with live HeV resulted in lower levels of cell–cell fusion in PaKiT03 cells, as well as sharp differences in host gene expression profiles [78], suggesting multiple host-specific factors involved in the regulation of HNV-induced syncytium formation, and further highlighting the need for comparative studies.

## 3. Mapping Syncytium Formation In Vivo

### 3.1. Knowledge from Other Paramyxoviruses

The observation of syncytia in vivo following paramyxovirus infections dates back to the beginning of the 20th century, when multinucleated cells were described in lymphoid and epithelial tissues of human and non-human primate MeV cases [83,84]. The role of syncytia in MeV pathogenesis seems widely acknowledged, as syncytia are frequently mentioned in reviews on the topic [28,29]. Such reviews are largely based on old anatomical pathology studies that often report syncytia [84,85], and include very few recent studies [86]. Syncytia are also observed following respiratory syncytial virus (RSV) infections, a virus formerly considered a paramyxovirus but recently placed in the closely related *Pneumoviridae* family [87]. RSV was named after observing syncytia induced by the virus in human cell culture [88], and syncytia were later identified as a characteristic of pulmonary lesions in RSV infections [9,89].

**Figure 3 viruses-13-01755-f003:**
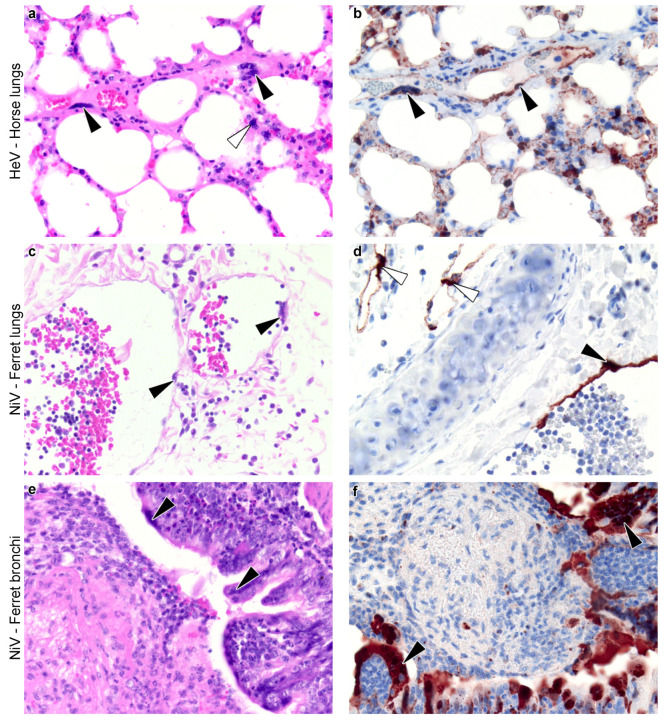
Henipavirus-induced syncytia in vivo. (**a**–**f**) Histological sections of lungs of animals experimentally infected with henipaviruses. Slides were stained with haematoxylin and eosin (**a**,**c**,**e**) and by immunohistochemistry for henipavirus nucleoprotein antigen (**b**,**d**,**f**) (brown color). Horse lungs (**a**,**b**, magnification x40) show syncytia of the blood vessel endothelium (black arrowheads) and associated with the alveolar walls (white arrowhead). Ferret lungs (**c**,**d**, magnification x20) show syncytia of the blood vessel endothelium (black arrowheads); in these images, the endothelium has fused with attenuation and sloughing, leaving regions of the vessel wall denuded. Virus has also infected the endothelium of lymphatic vessels (white arrowheads). Ferret bronchi (**e**,**f**, magnification x20) show syncytia of the bronchial epithelium (black arrowheads), some of which form into stalked structures; in these images, the bronchus is filled with inflammatory and cellular debris, transferred from elsewhere in the airways. These pictures were prepared specifically for this review from material taken from past studies ((**a**,**b**) [90]; (**c**,**f**) [91]).

In vivo, the occurrence and intensity of paramyxovirus-induced syncytium formation varies among tissues, hosts, and viruses. While MeV infections induce the formation of large syncytia in lymphoid and epithelial tissues, sometimes encompassing more than 30 nuclei, [28], syncytia do not seem to occur in nervous tissues [19]. MeV can nevertheless infect the brain, and neurons support MeV replication and transmission without cell–cell fusion [19]. The absence of syncytia in nervous tissues following MeV infection contrasts with canine distemper virus (CDV) infections, a closely-related paramyxovirus restricted to *Carnivora* hosts. CDV was found to induce syncytia in nervous tissues of domestic dogs (*Canis lupus familiaris*) [75]. Within nervous tissues, syncytia were more frequent in the meninges and white matter than in the gray matter, and were generally of small size (2–3 nuclei). Similarly to MeV, CDV-induced syncytia were also frequent and large in lymph nodes, often including more than 10 nuclei. However, syncytia were rare in the spleen, thymus, and lungs [75]. While experimental studies of CDV have mostly been restricted to domestic dogs, syncytia following natural infections have also been anecdotally reported across various *Carnivora* species such as foxes (*Vulpus* spp.), African wild dogs (*Lycaon pictus*), martens (*Martes* spp.), and raccoons (*Procyon* spp.) [92].

Overall, paramyxoviruses and pneumoviruses are interesting models to investigate the clinical and diagnostic relevance of syncytium formation. However, while MeV, CDV, and RSV have been relatively well studied, none represent an ideal model with which to study the distribution of syncytia across host and tissues due to the restriction of their host range (primates for MeV and *Carnivora* for CDV) or tissue tropism (respiratory tract for RSV). In contrast, HNVs have wide host range and tissue tropism, in line with their zoonotic potential [55], and thus provide fertile ground from which to investigate the differences in syncytium formation across host species and across host tissues.

### 3.2. Meta-Analysis of Henipavirus-Induced Syncytium Occurrence

#### 3.2.1. Data Availability

Based on a systematic review of the literature (detailed in the Appendix A), we built an extensive dataset compiling data on histology and viral material detection (viral genome detected by polymerase chain reaction (PCR) or viral proteins by immunohistochemistry (IHC)) from 91 publications reporting data from HNV infections, corresponding to more than 1200 individuals of 18 animal species and 2 modified animal models (Appendix A). We recovered syncytium data (i.e., explicit mention of presence or absence of syncytia) from 42 publications focusing on HeV or NiV. Among NiV, two clades are considered: the Malaysian (NiV-M) clade and the Bengali (NiV-B) clade. Those 42 studies correspond to approximately 150 individuals from 10 species (Figure 4): African green monkey (*Chlorocebus sabaeus*), domestic pig (*Sus scrofa domesticus*), domestic dog, domestic cat (*Felis catus*), domestic ferret (*Mustela putorius furo*), Guinea pig (*Cavia porcellus*), golden hamster (*Mesocricetus auratus*), horse (*Equus ferus caballus*), chicken (*Gallus gallus domesticus*), and humanized house mouse (*Mus musculus* with human lung xenograft). All the other studies were excluded from further analyses because they did not explicitly mention whether tissues samples were screened for syncytia or not, including the only two that considered CedV [34,38]. Several studies were conducted on bats and did not report any lesions except vasculitis in grey-headed flying foxes (*Pteropus poliocephalus*) exposed to HeV [35,93,94]. Among the 42 publications reporting syncytium data, a subset of 33 publications included explicit proportions of individuals presenting syncytia, 36 publications detailed which tissue(s) were examined, and 29 publications gave information on time of sample collection relative to infection. Only 23 publications fulfilled all three of these criteria, representing data from six host species (Appendix A). When syncytia were reported, they were usually reported as presence data only (with absence data not explicitly reported), and generally did not include any quantification of syncytia (e.g., number of nuclei included in syncytia or distribution of syncytium size), limiting the potential for quantitative inference from these data (but see [45,95]). This also limits the potential for cross-comparisons to in vitro studies that consider both the presence and quantification of syncytia.

Data availability on syncytium occurrence is heterogeneous across viruses, hosts, and tissues. Only four host species had data available for at least two viruses, enabling comparisons (Figure 4). Notably, only monkeys and hamsters had data available for all three of HeV, NiV-M, and NiV-B. When syncytium data were reported, the respiratory tract (in particular the lungs) was usually included, and data were also often available from nervous and lymphatic systems. In contrast, data from the reproductive and digestive systems were rarely reported, despite some evidence of syncytia from these tissues. Within an organ system, different organs were collected across studies (e.g., different lymph nodes, different structures from the brain, etc.); the resolution at which data are reported is also heterogeneous (e.g., “brain” versus specific brain structures; Appendix A). Experimental procedures also vary across studies, including eleven different virus inoculation routes, doses ranging from 68 to $5\times 10^{8}$ TCID_50_, and sample collection from 2 to 32 days post infection (Appendix A). All of these differences across study designs present challenges for comparison and synthesis of syncytium data.

#### 3.2.2. Syncytium Distribution across Tissues, Hosts, and Viruses

Syncytia were observed in all of the 10 host species for which syncytium occurrence data are available, and affect a wide range of tissues from the respiratory, digestive, lymphatic, nervous, cardiovascular, urinary, and reproductive systems (Figure 4). In particular, for almost all explored virus–host combinations, syncytia were detected in the lungs of a high proportion of individuals (60–100 %), even in the absence of gross lesions at necropsy (Appendix A). The sole exception was in Guinea pigs infected with NiV-M, where syncytia and other lesions were rare in the lungs despite the presence of low levels of viral antigens [45]. Across all host–virus combinations, syncytia were also often detected in the spleen and, to a lesser extent, kidneys (Appendix A). Syncytia have not been reported from rabbits and rats exposed to HeV [96], nor Egyptian fruit bats exposed to NiV-B [35], but those species were only considered in one study each, and appear not to be susceptible to the virus they received.

Besides the general high prevalences of syncytia in the lungs, spleen, and kidneys, the data do not highlight any other clear and consistent patterns across host and virus species, potentially because of the heterogeneity in data collection and reporting. For instance, syncytia are rarely observed in nervous tissues. This disparity could represent real biological differences, or be a consequence of varying methodologies between studies. For instance, Torres-Velez et al., 2008 reports that, in Guinea pigs infected with NiV-M, syncytia were present in the meninges specifically (structure composed primarily of fibroblasts and endothelial cells [97]), and it is not clear whether syncytia were also present in the neuroparenchyma and ependyma of the same individuals (Appendix A) [45]. Syncytium distribution in the brain may thus be heterogeneous, and syncytia may be missed if screening is not extensive. Most of the other studies focus on other parts of the brain, or do not report which parts of the brain were screened for syncytia, preventing comparison across studies. In addition, Guinea pigs were subjected to different inoculation routes (Appendix A) and sampling time (Appendix A). It is thus not possible to infer whether this observed difference is due to the viruses, inoculation routes and/or doses, sample collection protocols, or data reporting procedures.

Comparative studies, including data obtained from different experimental conditions but following the same protocols, could help address this issue. The available data do not reveal any detectable effects of inoculation route and dose on syncytium occurrence. For instance, syncytia were detected in respiratory and lymphoid tissues, but no other tissues, of monkeys infected via intranasal exposure of $2\times 10^{3}$ or $2\times 10^{4}$ PFU or via aerosol exposure to approximately $1\times 10^{3}$ or $1\times 10^{4}$ PFU of NiV-B [98,99]. A similar pattern was reported from monkeys exposed via intratracheal exposure to $2.5\times 10^{3}$ to $6.5\times 10^{4}$ PFU of NiV-M [100]. Similarly, Williamson et al. 2001 reported no dose effects on syncytia formed by endothelial cells in the kidney, bladder, and lungs but not in the brain of a hamster (with exposure doses of $3\times 10^{4}$ and $5\times 10^{4}$ TCID_50_ of HeV, although viral inocula with different laboratory histories were used) [101]. Other studies directly comparing inoculation route or dose do not include enough data to explore potential effects on syncytium formation or occurrence [91,95,102,103]. Between viruses, NiV-B appears to be more pathogenic than NiV-M in monkeys, with infected tissues presenting more viral antigens and more lesions, including syncytia [104]. One study reported that NiV-B progression was slower than NiV-M in hamsters, but did not include any in vivo syncytium data [76]; another one reported no detectable difference between NiV-M and NiV-B-induced lesions of the respiratory tract of hamsters [105].

Within a given tissue, when syncytia were observed, they were often in vascular endothelial or respiratory epithelial cells (Table 2). In particular, syncytia were observed at the epithelium of the nasal cavity and the lungs [105,106,107,108]. Syncytia were also observed, although more rarely, among epithelial cells of the tonsil [109] and the uterus [45]. More anecdotally, syncytia were reported from smooth muscle cells of blood vessels and macrophages [110]. Syncytia formed from neurons were explicitly reported only in hamsters [111] and chicken embryos [112], despite neurons being susceptible to infection in other species too [101,113].

A small number of studies present data as a function of time since infection, allowing the temporal pattern of syncytium occurrence to be examined (Appendix A). Syncytia were detected as early as two days post-infection (the earliest time-point available) in several conditions. In hamsters, syncytia were detected at two days post-infection in the nasal cavity and respiratory tract following NiV-M infection, and in the respiratory tract following NiV-B infection (Appendix A). In Guinea pigs infected with NiV-M, syncytia were detected at two days post-infection in the lymphatic system, but only at four days post-infection in the urinary tract (kidney and bladder) and reproductive system (ovary and uterus), and at seven days post-infection in the respiratory and gastro-intestinal tracts (Appendix A). From most of the other studies, the data do not allow the detection of temporal patterns in syncytium formation following HNV infection, because syncytia were either detected or not in a given tissue for the whole sampling period. This could be due to sampling often being restricted to a short period (up to 12 days for the majority of the studies). In particular, only three studies included data after 15 days post-infection, revealing that syncytia could still be detected in several tissues at least 28 days post-infection in Guinea pigs infected with HeV and 32 days post-infection in monkeys infected with NiV-M (Appendix A). In contrast, syncytia were not detected anymore in domestic pigs after 20 days following infection with HeV, suggesting that syncytia regressed between 7 and 20 days post-infection, but the temporal resolution of the data does not inform on the exact timing (Appendix A). However, some of the studies describe an increasing number of syncytia in the days following infection [114], and earlier appearance of syncytia in the lungs of hamsters infected with HeV compared to NiV-M [115], suggesting that data of higher resolution (notably including syncytium counts or measures of syncytium sizes, and examination of tissues across a fixed set of days post-infection) could potentially reveal temporal dynamics in syncytium formation.

Two studies reported lesion scores, providing a semi-quantitative metric of the extent of syncytium formation [45,110]. Two other studies specifically scored the extent of syncytium formation, but this was done only for lung samples and concerned very small sample sizes (1 to 4 individuals per experimental condition) [95,110]. These data cannot be compared quantitatively between studies as scoring criteria may vary. However, within a study, they provide information on the heterogeneous intensity of syncytium formation and associated lesions across tissues and individuals (Figure 5). For instance, in Guinea pigs at 4 days post-infection with NiV-M, syncytium formation, together with inflammation and necrosis, were particularly intense in the spleen, moderate to intense in the lymph nodes, bladder, ovaries, and uterus, and rare in the lungs, gastro-intestinal tract, and kidney [45]. In contrast, for Guinea pigs at 7–13 days post-infection with HeV, syncytium formation was more intense in the spleen and lungs [108]. Again, these studies should be compared with caution as, for instance, different sampling timings and notation criteria were used.

Syncytia were rarely observed in the absence of measurable viral proteins or viral genome. Though the temporal data are sparse, some host–virus pairs show evidence that viral antigens increase first, and syncytia follow (Figure 6). However, not all infected tissues exhibited syncytia (Figure 6). For instance, despite the overall low prevalence of syncytia, HNV antigens are commonly found in the nervous system of various species infected with HeV, NiV-M, or NiV-B including monkeys [121], cats [102], ferrets [122,123], Guinea pigs [101], and hamsters [124]. Infected tissues generally present lesions typical of inflammation, such as the infiltration of lymphocytes and other inflammatory cells, and necrosis, haemorrhage, and fibrin deposition; this is reported with or without the presence of syncytia. However, endothelial syncytia also occur independently from any other lesion, as noted in human NiV-M cases [125].

## 4. Discussion

### 4.1. Identified Patterns

Our review highlights that syncytia are widely distributed in animals infected by pathogenic HNVs (Figure 1a). Syncytia were particularly reported in the lungs, spleen, and kidney of infected animals. However, syncytium detection varies across tissues, hosts, and experimental conditions, which parallels in vitro observations. These variations may be driven by methodological constraints, such as lower detection probability in tissues where syncytium formation is rare, suggesting that the intensity of syncytium formation varies between tissues as well. Without specifically designed comparative studies, inferring biological differences from data collected using different protocols requires an accurate description of the observed syncytium distribution, together with an exhaustive description of the methods.

For instance, while some studies conducted on monkeys report data at a relatively fine scale, the relatively rare reports of syncytia in the nervous system does not exclude the possibility of syncytia being present but missed instead. This is exacerbated by the fact that, across all host species, the only studies explicitly screening the meninges specifically both reported syncytia in the nervous system [45,112]. Unfortunately, no other study of monkey infections explicitly mentions whether the meninges were screened for syncytia. Neuronal syncytia were also reported in hamsters [111], but they were noted as rare, suggesting that they could easily be missed. The fact that neurons formed syncytia in chicken embryos should be interpreted with caution as the spatial and temporal dynamics of ephrin expression are still poorly described [126]; in particular, as ephrins are involved in the development of various tissues, including nervous tissues [126], one could expect particularly high expression levels in embryonic neurons. Nevertheless, considering that syncytia form in neurons in vitro (Table 1), that neurons are susceptible to infection in vivo [101,112], and that HNV infections are associated with neurological disorders in many hosts (Table 3), it would not be surprising for syncytia to occur in the nervous system of various species in vivo. In contrast, the presence of syncytia in some tissues, notably the kidney, spleen, and reproductive system, does not appear to be associated with specific clinical signs. Blood chemistry analyses have revealed evidence of kidney damage in ferrets and monkeys infected with NiV-M [121,123], but effects may vary across species [127].

Quantitative data from one paper suggest that the extent of syncytium formation (and other lesions) is correlated with viral antigen quantity measured by IHC, which is a proxy of infection intensity [45]. This pattern cannot be extrapolated beyond this study, as no others explicitly reported both syncytium formation and infection intensity scores. In addition, within a given tissue, syncytia are generally detected concomitantly or just after viral antigens, reinforcing the potential causal link between infection and syncytium formation. Experimental approaches would be necessary to assess whether this correlation is due to a positive effect of syncytium formation on viral replication, or the other way around. For instance, in vitro, light shaking of cell culture plates might prevent cell–cell fusion, hence disentangling the contribution of free-virion transmission versus direct cell-to-cell transmission to viral fitness [13]. This would represent a first step into our investigation of the contribution of syncytium formation to pathogenesis in vivo, which depends on virus replication, spread, and associated lesions. However, some other studies reported infected tissues without any syncytia. Why some tissues present syncytia and others do not, and the implications for virus fitness, remain to be elucidated. Host and tissue differences in ephrin expression and virus affinity for specific ephrins may contribute to a propensity for syncytium formation. For instance, both ephrin B2 and ephrin B3 are widely expressed in endothelial cells, and syncytia are observed in endothelial cells of all the species considered. However, ephrin B3 is expressed in the brain stem, but ephrin B2 is not, so the lower affinity of HeV, compared to NiV, for ephrin B3 could explain why syncytia were more often detected in the brain of Guinea pigs infected with NiV-M than HeV, although viral antigens and other lesions were detected in both cases [45,101]. Very few data are available on cathepsin expression across hosts, tissues, and cell types [126], limiting the exploration of a potential effect of cathepsin expression on syncytium formation.

Data on syncytium occurrence following natural infection are scarce. Observations from domestic pigs affected by the 1998 Malaysian NiV outbreak align with laboratory observations: infection resulted in respiratory and neurological syndromes, and syncytia were detected in the respiratory epithelial cells, as well as in endothelial and smooth muscle cells of the cardiovascular and lymphatic systems [148]. Similarly, horses affected by the 1994 Australian HeV outbreak presented respiratory disorders and endothelial syncytia in the lungs [149]. Thus it is clear that syncytium formation in vivo is not limited to experimental infections, which are sometimes suspected to diverge from natural infections notably because of the high inoculation doses. In humans, HeV and NiV infections cause acute respiratory and neurological disorders, as in non-human primate models. Syncytia have been detected in human brain, kidney, and lymphoid samples, and were particularly present in blood vessels, following NiV infection [125], which aligns with observations from monkeys and other animal models. In contrast, HeV infections in humans were not associated with syncytia in the four cases for which samples were available [150]. In addition, HeV and NiV can cause late-onset or relapsed encephalitis in humans [125]. This aspect of HNV infections is difficult to study in animal models (notably because of the relatively short duration of infection experiments in animals), although it would be interesting to assess whether direct cell-to-cell transmission could contribute to chronic brain infections as it does for MeV [19].

### 4.2. Methodological Challenges of Syncytium Mapping

Meta-analyses can make important contributions to science by addressing questions that cannot be addressed with studies focusing on specific experimental conditions (in particular a specific host or virus species, or inoculation protocol) [151], or by increasing statistical power via increased sample sizes [152]. However, meta-analyses rely on standardized methods and detailed reports of data. Despite repeated calls for systematic data sharing over the last few decades [153], methods and data are rarely reported with enough detail to allow their comparison or integration.

For example, our meta-analysis was challenged by the lack of systematic reporting of syncytium screening and detection, despite a large majority of in vivo studies that included histological investigations. In particular, reported syncytium data are often limited to a brief verbal statement (e.g., “in this organ, lesions included syncytia”) or a picture of an illustrative histology slide. Similarly, quantitative data, such as viral loads, are often provided only in plots. These data are difficult to use for meta-analyses or novel statistical analyses as (1) data are often incomplete (i.e., not available for all the samples), (2) individual data from different analyses (e.g., viral load quantification and histology) cannot be combined, and (3) data extraction can be inaccurate (especially when only aggregated values, such as mean values or intervals, are reported).

Furthermore, ethical arguments highlight the importance of sharing data from studies involving animals [151]. To be useable, data should be shared in a raw format (i.e., individual data points instead of aggregated values) and accompanied by detailed metadata (e.g., individual identification, experimental treatment, etc.) [154]. A culture of consistent archiving of raw data, when available, alongside published studies, as is now standard in other scientific fields, would advance the study of virology and ensure maximum scientific gain from animal experiments.

In addition to data sharing, improved practices in data collection could enable further advances in histological investigations. Several arguments suggest that syncytia could be missed even when present. First, in a given organ, syncytium distribution can be heterogeneous, notably in complex tissues composed of numerous different structures (e.g., the brain). If syncytium screening is not exhaustive (i.e., not including all the different structures), then syncytium absence cannot be inferred from syncytium non-detection (i.e. absence of evidence is not evidence of absence). In such cases, structures screened for syncytium should be precisely described (e.g., which section of the brain, and how many slices of what dimensions). Second, the few studies including syncytium intensity scores reported that intensity could vary among tissues or individuals (Figure 5), and that syncytia could be present but rare in some samples. This implies that accurate syncytium detection requires extensive screening, i.e., complete, or repeated (screening X fields from Y slides) randomized spatial coverage. Statistical approaches that capitalize on such repeated sampling are widely used in species distribution modeling, and have already been shown to be beneficial when applied to histological investigations [155]. As random screening can be time consuming and inefficient, especially in tissues showing rare syncytia and for which a high number of fields and slides would have to be screened to reach reasonable levels of confidence in detection/non-detection data [156], another option could be to use two-step designs, similar the approaches used to detect rare species [157]. In the case of syncytium detection, the first step could consist of an extensive screening of samples for viral antigens (e.g., using IHC), and the second step would focus on syncytium detection within the infection foci previously identified. In addition to improving detection, such approaches also inform detection probability (by allowing the estimate of uncertainty in linked detection, e.g., if syncytia were not detected from a sample, what was the probability of syncytia being truly absent from this sample?) and can be used to improve process intensity estimation [158].

### 4.3. Comparison of Syncytium Occurrence Patterns In Vitro and In Vivo

Even though cell–cell fusion in vitro and in vivo are likely governed by the same underlying mechanisms, the additional factors at play in vivo and the different methods used to monitor cell–cell fusion in vitro and in vivo (Box 1) raise concerns about correlations between in vivo and in vitro data. Cell lines used in in vitro studies, while convenient and reproducible, are not necessarily representative of the natural pathogenic effects of HNV infections. For instance, NiV and HeV infections are characterized by severe encephalitis and pneumonia (Table 3), but in vitro studies seldom focus on neuronal or respiratory cells but instead on kidney (e.g., Vero, HEK293T, BHK) cells. Our meta-analysis, however, highlighted that syncytia are often formed in kidneys in vivo, and in endothelial cells in general. The absence of renal syndromes may thus be due to the fact that renal functions are particularly robust rather than being a consequence of tissue tropism. The correlation between in vitro and in vivo data is nevertheless imperfect. For instance, NiV and HeV can infect domestic pigs [106,159] but show no fusion in pig kidney (PK13) cells, which likely express low to null levels of ephrin [20]. Reciprocally, while CedV is non-pathogenic in vivo, it induces cell–cell fusion in Vero cells [38]. This specific discordance could be explained by the fact that Vero cells lack interferons [160], while the low pathogenicity of CedV in vivo has been attributed to its inability to suppress host interferon response (because it lacks the non-structural proteins V and W) [34,38]. Hence, interferons likely impact viral replication, but not directly cell–cell fusion itself.

The over-representation of syncytia formed from epithelial and endothelial cells, compared to other cell types, may be due to either particularly high exposure of those cells to virions (due to their localization alongside the routes of transit of viruses: in airways, vessels, and ducts), or a particularly high propensity of these cells to fuse (notably because of tight intercellular junctions). In contrast, interstitial cells might be less exposed to virions and/or less likely to form syncytia even if susceptible to infection, notably as many express ephrins, as is the case for fibroblasts (e.g., Vero cells; Table 1) and neuroglial cells (e.g., astrocytes and oligodendrocytes [125]). In the case of neurons, syncytia may be rarely reported in vivo despite frequent observations in vitro because of the inherent structure of nervous tissues, where cells are connected only at their synapses, which comprise a relatively low proportion of their membranes. In both cases, syncytium formation in mono-layer cell cultures in vitro might be facilitated by increased contact surfaces between cells compared to in vivo. In addition, epithelial and endothelial syncytia may be reported more often because these cells can easily be identified based on their location, relative to other cell types, leading to a reporting bias that is not necessarily representative of an actual biological pattern. Interestingly, cell–cell fusion can occur between cells of different types in vitro (e.g., HEK293T and Vero cells [66]). This has not been reported in vivo, potentially as a consequence of difficult cell identification, especially after cell–cell fusion. While in vitro cell type is chosen by the experimenter, cell-specific markers (e.g., by immunolabeling) could be used as part of histological analyses of samples collected in vivo to improve the identification of cell types involved in syncytium formation.

These examples illustrate that inferring the implications of syncytium formation for viral spread in vivo from in vitro studies should be done with caution, especially when using immortalized cell lines that present phenotypic variations from primary cell lines [161], or using mono-layer cell cultures. Nevertheless, immortalized cell lines remain excellent practical models with which to investigate the implications of various factors in membrane fusion in a controlled environment.

### 4.4. Knowns and Unknowns of Syncytium Formation and Future Directions

The wide occurrence of syncytia following HNV infections in vivo supports the idea that this phenotype is not a cell culture artifact, but rather a biologically relevant feature of HNVs. This observation reinforces the idea that cell–cell fusion might represent, or be associated with a process representing, an evolutionary benefit. However, experimental evidence is lacking, and we cannot exclude that syncytium formation is a by-product of other viral functions (e.g., virus–cell fusion). Beyond HNVs, the potential benefit of cell–cell fusion for viruses is also supported by the evolution of non-structural proteins in non-enveloped viruses inducing cell–cell, but not virus–cell, fusion [162]. The potential evolutionary benefit of syncytium formation could arise from enhanced ability to infect other cells, immune evasion, or the creation of multinucleated “virus factories” [10,11,25,163]. Experimental studies are needed to discriminate between these hypotheses. Paradoxically, cytopathic effects linked to syncytium formation might impair cell functioning, hence also decreasing virus replication [26,27,164,165]. In addition, cell–cell fusion might compete with virion release (e.g., if cell–cell fusion “consumes” viral glycoproteins), which is ultimately required for transmission between individuals [70], giving rise to a within- versus among-host spread trade-off. The comparison of mutants exhibiting different levels of fusogenicity would be particularly insightful to understand the mechanisms driving the relationship between virus fitness and syncytium formation [14,15,16]. In vitro studies can approximate the relative viral fitness of syncytia compared to mononucleated cells by considering the number of infectious virions released and/or secondary cellular infections caused. A more refined investigation of the correlation between syncytium size (i.e., number of cell nuclei) and viral fitness would be especially informative as this relationship might be non-monotonic (i.e., peaked) if there exists an optimal syncytium size for virion production. We thus encourage researchers studying HNVs to report detailed records of syncytium size distribution in addition to measures of virus-like particle budding (when using non-replicative artificial constructs), infectious virion release (when using live viruses in vitro), or viral load (when using live viruses in vivo). Such studies should include various virus–host combinations as syncytium formation is impacted by various viral and host factors. Studies focused on herpesviruses have notably revealed that cell–cell fusion phenotype can significantly differ across isolates of the same viral species [166]. As sequences of henipaviruses detected in domestic and wild species are becoming available [167,168,169], it is important to consider a diversity of henipavirus isolates in comparative studies to understand how cell–cell fusion phenotype varies across viruses and how it correlates with spillover risk.

A refined temporal and spatial description of syncytium occurrence in vivo could help decipher the role of cell–cell fusion in virus dissemination and organ colonization. For instance, much interest has focused on brain colonization by HNVs and its possible relation to development of encephalitis, but the contribution of cell–cell fusion to encephalitis remains elusive. In hamsters and domestic pigs infected with NiV-M, spatiotemporal patterns of lesions and detection of viral antigens lend support to colonization of the neurological system via axonal transport along cranial nerves, in particular the olfactory nerve [107,128]. Syncytia were observed on the epithelium of the nasal cavity, but their occurrence is not reported with a fine enough resolution to assess whether they could also play a role in virus dissemination in parallel to axonal pathways (e.g., by mapping the spatio-temporal progression of infection, as in [107], in parallel to syncytia). Disruption of the blood–brain barrier has also been suggested as an important pathway to brain colonisation by HeV and NiV [128,170]. As syncytia frequently form from endothelial cells across many organs, it is plausible that they contribute to the disruption of the blood–brain barrier. In vitro, the loss of impermeability of an endothelial cell culture correlated not with the beginning of viral replication, but with the appearance of syncytia and cytopathic effects [171]. Unfortunately, very few in vivo studies report syncytium data from brain samples.

In addition to correlative studies, experimental approaches can investigate the impact of various viral and cellular factors on syncytium formation. The identification of viral mutants supporting higher or lower levels of cell–cell fusion but conserved levels of virus–cell fusion (or vice-versa) would shed light on the viral factors driving cell–cell fusion specifically. More emphasis on quantitative measurements would also provide valuable information. For instance, measuring the impact of increased ephrin expression (using transfected cells [20]) or cathepsin activity (using enzymatic inhibitors [58,59,60]) on cell–cell fusion and virus–cell fusion could highlight unsuspected differences in the cellular requirements for both processes. Similarly measuring the impact of G, F, and ephrin expression ratios on the two processes could provide information on the stochiometry of the protein interactions. In particular, it remains unknown whether cell–cell fusion requires the involvement of more fusion complexes (i.e., G, F, and ephrin polymers) than virus–cell fusion does. Such experimental knowledge would be especially useful if complemented by studies on the natural expression levels of host and virus proteins using methods such as proteomics [78] or flow-virometry [172].

If syncytium formation impacts viral replication, then variations in susceptibility to syncytium formation across hosts and tissues may result in differences in viral fitness. Within a host, this can manifest as gradients of local fitness across tissues, with some permitting more viral replication than others. However, the tissues contributing to viral replication and to viral spread between tissues may be different. Further, the fitness impact of syncytium formation and, more practically, the dominant mode of viral transmission may vary across tissues. Syncytia and other cell-to-cell transmission mechanisms may enable the virus to access otherwise inaccessible tissues. For instance, direct cell-to-cell transmission modes may be especially advantageous at respiratory epithelial interfaces, where mucosal immunity creates an additional barrier to infection by extracellular viruses [11]. The dissemination of HNVs by binding, but not infecting, highly mobile leukocytes is an extreme example of viral replication and within-host dissemination independence [173].

Our review highlights that susceptibility to syncytium formation generally, but not always, correlates qualitatively in vitro and in vivo. The occasional exceptions are not surprising considering the inherent complexity of in vivo systems (Box 1). It is critical to identify the factors modulating susceptibility to syncytium formation in order to design in vitro assays that can predict pathogenesis (e.g., does consideration of immune factors allow the reconciliation of in vitro and in vivo patterns?). In addition, as syncytium-forming viruses have been proposed as a strategy for cancer treatments [174], better understanding of the drivers and consequences of syncytium formation can assist in the development of oncolytic virotherapy. Organoids and organs-on-chips present another promising avenue for investigation, as they occupy the middle ground between cell culture and in vivo experiments [175,176]. These setups notably allow researchers to recover features classically associated with in vivo settings, such as complex three-dimensional spatial structures involving heterogeneous cell types. They also enable ethical investigations with levels of detail classically associated with in vitro settings such as higher time resolutions, higher numbers of specimens, and more experimental conditions. In particular, high temporal resolution would permit tracking the dynamics of syncytium and virus dissemination in space and time. Organoids have recently been used to study RSV pathogenesis in airways, and successfully recovered various features of in vivo infection, including syncytium formation and the recruitment of neutrophils when co-cultured with the organoids [177].

## 5. Conclusions

HNV-induced syncytium formation occurs widely across hosts, tissues, and virus species, both in vitro and in vivo. However, susceptibility to syncytium formation varies independently from susceptibility to infection. The host and virus factors driving these variations and their impact on viral fitness and pathogenicity remain largely unknown. Considering cell–cell fusion and free-virion release together in an integrated and quantitative framework could yield considerable progress in studying the pathogenesis of HNVs and other syncytium-forming viruses. Such insights would enrich our understanding of the drivers of viral fitness more broadly. From the perspective of monitoring emerging infectious diseases, if syncytium formation can be used as a proxy of viral fitness or pathogenicity in some conditions, then cell–cell fusion assays—which can be conducted in BSL2 using only binding and fusion protein sequences—could enable rapid and ethical zoonotic risk assessment of novel paramyxoviruses. The wider use of such assays, in appropriate conditions, would contribute to a framework oriented towards the functional, rather than the genetic, characterization of emerging viruses [178].

## Figures and Tables

**Figure 1 viruses-13-01755-f001:**
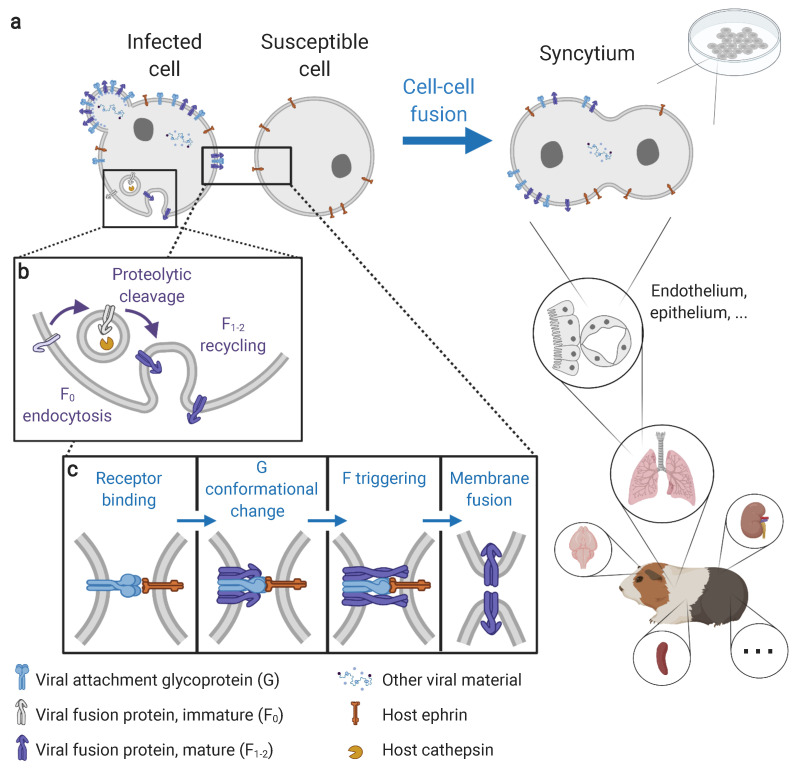
Molecular mechanisms and cellular manifestations of henipavirus-induced syncytium formation. (**a**) Syncytia are multinucleated cells formed from the fusion of an infected cell with a susceptible cell. Syncytia can be observed both in vitro (in cell culture) and in vivo (here in a Guinea pig). In vivo, syncytia are observed in a wide range of tissues, including the vascular, respiratory, nervous, lymphatic, and urinary systems, particularly (but not exclusively) at endothelial and epithelial interfaces. Syncytia can occur both between infected and non-infected cells, and between infected cells. (**b**) Fusion protein maturation via the proteolytic activation by host cathepsins in endosomes. (**c**) Membrane fusion cascade via the interactions between viral envelope proteins (G and F) and host receptors (ephrins). Membrane fusion is a pH-independent process using an attachment-mediated triggering mechanism of the fusion protein. Steps preceding membrane fusion (in particular infection, viral replication, and viral protein expression and egress) are detailed in Howley and Knipe 2020 [46]. Visual created using BioRender.com on 21 August 2021.

**Figure 2 viruses-13-01755-f002:**
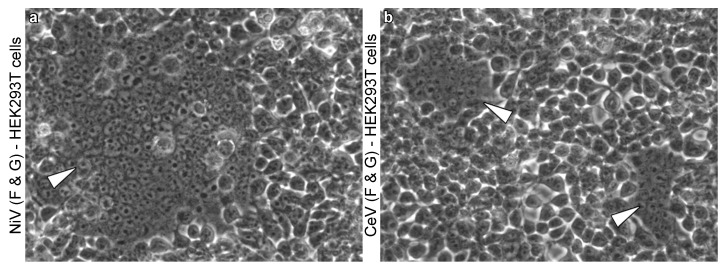
Henipavirus-induced syncytia in vitro. (**a**,**b**) HEK293T cells transfected with the G and F proteins of NiV (**a**) and CedV (**b**) in vitro. White arrowheads point towards syncytia ((**a**): one large syncytium encompassing more than 100 nuclei; (**b**): two syncytia encompassing approximately 20–30 nuclei each). Magnification x200. Detailed methods are available in Yeo et al. 2021 [48].

**Figure 4 viruses-13-01755-f004:**
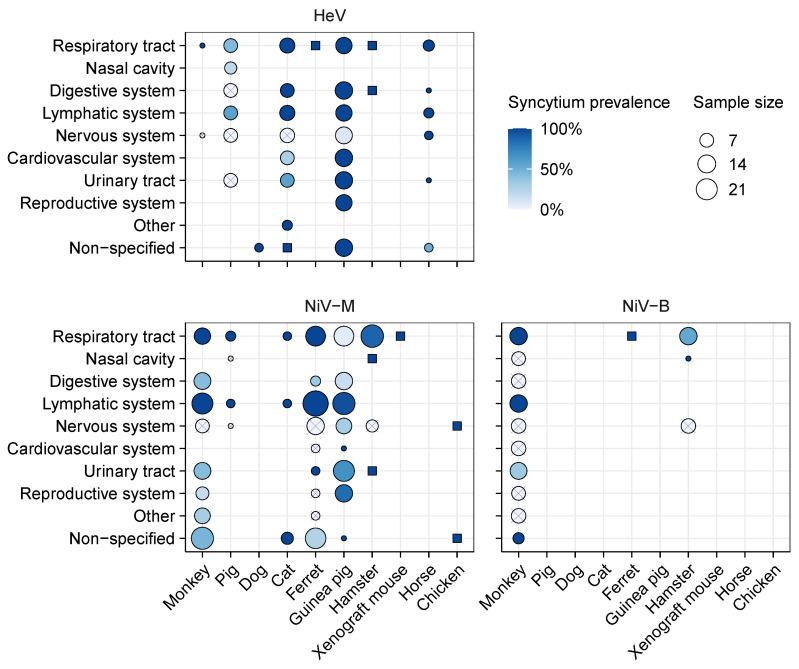
Proportion of individuals presenting syncytia in different tissues following experimental henipavirus infection in vivo. Crossed circles indicate that no syncytia were observed; dark blue squares indicate that syncytia were observed but the exact proportion of individuals is unknown; blank spaces indicate that no data are available. Data obtained using different inoculation routes or doses and collected at different times post-infection were merged together. Available immunohistochemistry (IHC) data and the proportion of individuals presenting viral antigens are plotted in Appendix A for comparison. Data were extracted from the studies listed in Appendix A.

**Figure 5 viruses-13-01755-f005:**
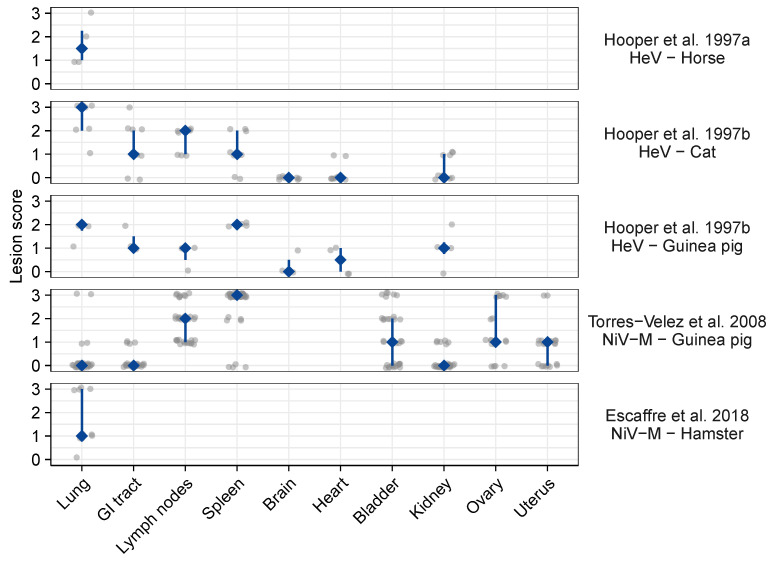
Scoring of histological lesions (including but not restricted to syncytia) following experimental henipavirus infection in vivo. Grey dots represent individual data points (horizontally jittered to avoid overplotting); dark blue diamonds and vertical bars represent the corresponding median and interquartile ranges; blank spaces indicate that no data are available. GI tract: gastro-intestinal tract. Individuals received different doses and samples were collected from 2 to 17 days post-infection. Lesion scores were based on observations of hematoxylin-and-eosin stained slides, and attributed as following. Hooper et al. 1997a [110]: presence of syncytia was noted on a comparative scale from 0 to 3. Hooper et al. 1997b [108]: presence of lesions was on a comparative scale from 0 to 3; lesions were syncytial cells, particularly but not exclusively in vascular endothelium, edema, cellular infiltration, and necrosis. Torres-Velez et al. 2008 [45]: 0, no abnormal findings; 1, minimal changes consisting of focal inflammation with some necrosis and rare syncytial cells; 2, moderate changes consisting of necrosis that can be focally extensive with or without inflammation, syncytial cells are usually prominent; 3, severe inflammation, extensive necrosis, hemorrhage, and numerous syncytial cells. Escaffre et al. 2018 [95]: 0, no detectable syncytia; 1, rare (1–2 syncytia/section); 2, occasional (3–5 syncytia/section); 3, frequent (5 or more syncytia/section).

**Figure 6 viruses-13-01755-f006:**
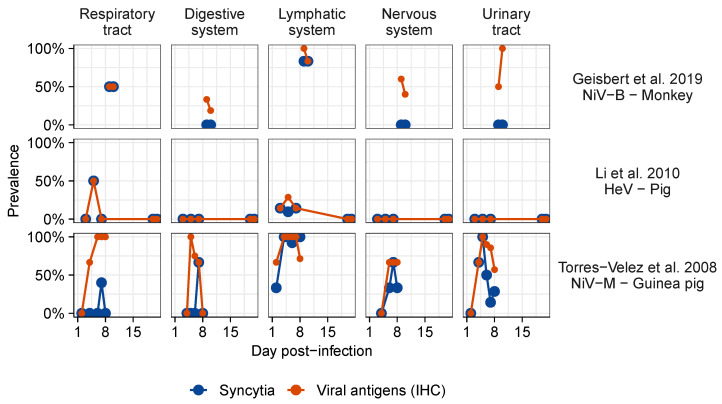
Proportion of individuals presenting syncytia and viral antigens detected by immunohistochemistry (IHC) over time post-infection following experimental henipavirus infection in vivo. Data from the most complete studies for each virus were selected. Data were extracted from [45,98,106]. The complete dataset showing the temporal patterns of syncytium detection is represented in Appendix A.

**Table 1 viruses-13-01755-t001:** Occurrence of henipavirus-induced syncytia in vitro. A “+” indicates that syncytia were observed, and a “–” that syncytia were not observed, following either live infection (“*live*”) or viral glycoprotein transfection (“*transf.*”); blank cells indicate that no data are available; non-exhaustive list. There are no data from KuV and MojV live infection experiments as these viruses have not been isolated but sequenced only [39,71].

Cell Line	HeV		NiV		CedV		KuV	MojV	References
	live	transf.	live	transf.	live	transf.	transf.	transf.	
Vero (epithelial)	+	+	+	+	+	+	-	+	[20,79,80]
HEK293T (epithelial)		+		+		+	+	+	[20,79]
A549 (epithelial)		+		+		+	+	+	[70,79]
Raji B (lymphoblast)				-					[20]
PK13 (epithelial)		–		-					[20,31]
BHK (fibroblast)	+	+	+	+	+	+	–	+	[79,80]
CHO (epithelial)		–		–		–		–	[48,50,79]
L2 (epithelial)		+				+		+	[79]
Rat2 (fibroblast)		+				+		+	[79]
EidNi (epithelial)							+		[80]
NHBE (epithelial)	+		+						[81]
SAEC (epithelial)	+		+						[81]
hOE (epithelial)	+		+						[82]
HMVEC (endothelial)				+					[20]

Vero: monkey renal epithelial cells; HEK293T: human renal epithelial cells; A549: human pulmonary epithelial cells; Raji B: human lymphoblast cells; PK13: porcine renal epithelial cells; BHK: baby hamster fibroblasts; CHO: Chinese hamster ovary epithelial cells; L2: rat pulmonary epithelial cells; Rat2: rat fibroblast; EidNi: *Eidolon* bat renal epithelial cells; NHBE: human bronchi epithelial cells; SAEC: human small airway epithelial cells; hOE: human olfactory epithelial cells; HMVEC: human microvascular endothelial cells.

**Table 2 viruses-13-01755-t002:** Syncytium detection across cell types and hosts following experimental henipavirus infection in vivo. A “+” indicates that syncytia were reported at least once; blank spaces indicate that no data are available.

		Hamster	Cat	Horse	Guinea Pig	Pig	Dog	Ferret	Monkey	Chicken
HeV	Endothelial			+ [110]	+ [108]				+ [115]	
	Epithelial					+ [106]	+ [109]	+ [94]		
	Macrophage			+ [110]						
	Neuron									
	Smooth muscle			+ [110]						
NiV-M	Endothelial	+ [116]	+ [102,117]					+ [118,119]	+ [120]	
	Epithelial	+ [105]	+ [102,117]		+ [45]			+ [119]		
	Macrophage									
	Neuron									+ [112]
	Smooth muscle									
NiV-B	Endothelial							+ [119]		
	Epithelial	+ [105]						+ [119]		
	Macrophage									
	Neuron									
	Smooth muscle									

**Table 3 viruses-13-01755-t003:** Clinical signs resulting from experimental henipavirus infections. Blank cells indicate that no data are available.

	HeV	NiV-M	NiV-B
Monkey	Respiratory disorders [115]	Respiratory and neurological disorders [100,104,113]	Respiratory disorders [98,99,104]
Pig		Respiratory and neurological disorders [128,129,130,131]	No clinical sign [132]
Dog	Mild non-specific symptoms [109]		
Cat	Respiratory disorders [96]	Respiratory disorders, no neurological disorders [102,117]	
Ferret	Neurological disorders [133]	Respiratory and neurological disorders [119,134]	Respiratory and neurological disorder [119,134]
Guinea pig	Little to no respiratory disorder, neurological disorders [101,135]	Respiratory and neurological disorders [45,136]	
Hamster	Respiratory and neurological disorders [114,124,137,138]	Respiratory and neurological disorders [76,95,107,116,127,137,138,139,140,141,142,143]	Respiratory and neurological disorders [76,138]
Mouse	Neurological disorders [144]	No clinical sign [116,145]	No clinical sign [145]
Horse	Respiratory disorders [146] (neurological disorders also noted from field cases [147])		
Fruit bat	No clinical sign [93,135]	No clinical sign [136]	

## Data Availability

Data and code to reproduce the analyses and figures presented in this manuscript are available online at https://doi.org/10.5281/zenodo.5338802 [194].

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
