# Peer review of "Drivers and Distribution of Henipavirus-Induced Syncytia: What Do We Know?"

_viruses, 2021, doi:10.3390/v13091755_

Round 1
Reviewer 1 Report
Gamble et al. summarize the current knowledge on syncytium formation in vitro and in vivo with focus on henipaviruses and highlight the data gaps that affect the overall understanding of the relevance of syncytium formation for virus pathogenesis. Authors include the most relevant literature on henipavirus syncytium formation and describe the problems that arise in comparing especially with respect to the data on in vivo syncytium formation. They conclude that for using syncytium formation as a proxy for virus pathogenicity, it would be necessary to fill the gaps, publish data more precisely with respect to what was e.g. not detected in vivo studies, and adjust experimental conditions for in vitro experimental approaches for better comparison.
Overall, the review is quite extensive (especially if one considers the amount of supplemental material on top) and detailed, but is written well and comprehensible.
However, I have very few remarks:
> Authors start the Abstract with paramyxoviruses, then quickly come to henipaviruses which they mainly focus on. Of course, in the introduction the overall relevance of syncytium formation reported for various virus families has to be presented. Then, however, throughout the manuscript, it would be preferable to stay with henipaviruses (at most include references to paramyxoviruses) and not include bits of information for other viruses since this makes the review even more complex. As an example: Reference to coronaviruses in ll. 127,128.
> ll. 44 -46: Isn’t there a contradiction in this sentence? “… the virion assembly and budding steps are skipped…potentially accelerating virion production…”?
> Figure 1: Please change the word “exocytosis” to “recycling” since exocytosis is a different cellular process. Authors describe the process correct in l. 110.
> Maybe authors could add/discuss the fact, the quantitation of fusion in vitro cannot be performed with all relevant cell lines since some cells cannot be transfected (include in Box 1 as a challenge?).
> Could authors give a definition from what quantity of cell nuclei included we speak of a syncytium? Is there a minimum number? In ll.173/174 authors state “small size with 2-3 nuclei”? Is there a general definition out?
> Figure 3: Please add from which (published?) study/studies these picture are taken.
> Figure 4: What is the meaning of the included squares?
> Figure 5: Even though I was able to get the relevant information, I think the Map is difficult to read (space between the various cells is small). Could this be converted in a table?
> Figure 6: Please adjust order in Figure itself with Figure Legend (e.g. start both times with Hooper et al., 1997b).
Minor points to consider:
> l. 2: In the Abstract authors state “other important viruses”. Important in what respect? Use a different word?
> l. 33: panencephalitis?
> l. 73: word missing: … to understand the of syncytium…?
Figure legend to Figure 6: Spellings: Individuals received different doses and were (?) samples were collected from 2 to 17 days post infections (infection ?).
Figure 7: Prevalence in %? 1 equals 100%?
> l. 370: additional word? … 1994 HeV Australian HeV outbreak
> ll. 471 – 474. Can you please edit this sentence? This sentence is awfully long and nested.
> Funding: H.A.C should be H.C.A?
Author Response
Gamble et al. summarize the current knowledge on syncytium formation in vitro and in vivo with focus on henipaviruses and highlight the data gaps that affect the overall understanding of the relevance of syncytium formation for virus pathogenesis. Authors include the most relevant literature on henipavirus syncytium formation and describe the problems that arise in comparing especially with respect to the data on in vivo syncytium formation. They conclude that for using syncytium formation as a proxy for virus pathogenicity, it would be necessary to fill the gaps, publish data more precisely with respect to what was e.g. not detected in vivo studies, and adjust experimental conditions for in vitro experimental approaches for better comparison.
Overall, the review is quite extensive (especially if one considers the amount of supplemental material on top) and detailed, but is written well and comprehensible.
*** Authors’ response - We thank the reviewer for their positive evaluation of our manuscript and their constructive comments.
However, I have very few remarks:
> Authors start the Abstract with paramyxoviruses, then quickly come to henipaviruses which they mainly focus on. Of course, in the introduction the overall relevance of syncytium formation reported for various virus families has to be presented. Then, however, throughout the manuscript, it would be preferable to stay with henipaviruses (at most include references to paramyxoviruses) and not include bits of information for other viruses since this makes the review even more complex. As an example: Reference to coronaviruses in ll. 127,128.
*** Authors’ response - We have removed this mention of coronaviruses. The sections “Insights on syncytium formation from in vitro studies” and “Mapping syncytium formation in vivo” now focus on paramyxoviruses exclusively. However, we have kept mentions of other viruses in the Introduction and Discussion to compare knowledge availability between henipaviruses and other viruses (more studied, like HIVs and herpesviruses) and discuss the contribution of syncytium formation to virus fitness and virulence. In addition, Reviewer 2 suggested including additional mentions of other viruses.
> ll. 44 -46: Isn’t there a contradiction in this sentence? “… the virion assembly and budding steps are skipped…potentially accelerating virion production…”?
*** Authors’ response - Thank you for pointing out this contradiction. We have corrected this by rewording the second part of the sentence. It now reads “during direct cell-to-cell transmission, the virion assembly and budding steps are skipped, potentially accelerating the dissemination of viral material” (lines 47-49 of the revised PDF).
> Figure 1: Please change the word “exocytosis” to “recycling” since exocytosis is a different cellular process. Authors describe the process correct in l. 110.
*** Authors’ response - We have modified Figure 1 accordingly.
> Maybe authors could add/discuss the fact, the quantitation of fusion in vitro cannot be performed with all relevant cell lines since some cells cannot be transfected (include in Box 1 as a challenge?).
*** Authors’ response - This is a very good point. We have added a sentence on this topic in Box 1: “This approach is limited to cell lines that are susceptible to transfection, and is thus challenging (and rarely implemented) in many relevant cell types.” (in the “Inducing cell-cell fusion in vitro” paragraph).
> Could authors give a definition from what quantity of cell nuclei included we speak of a syncytium? Is there a minimum number? In ll.173/174 authors state “small size with 2-3 nuclei”? Is there a general definition out?
*** Authors’ response - Unfortunately, there is no general definition about the number of nuclei necessary for a cell to be considered a syncytia. This adds another challenge to the direct comparison of intensities of syncytium formation between in vitro and in vivo conditions. Some in vivo studies mention syncytia with 2 nuclei (e.g., Summer et al. 1985, https://doi.org/10.1016/0021-9975(85)90047-7), while many in vitro studies using manual counting of syncytia (in opposition to reporter genes) only consider cells with four or more nuclei (e.g., Stone et al. 2006, https://doi.org/10.1371/journal.ppat.1005445, Bradel-Tretheway et al. 2019, https://doi.org/10.1128/JVI.00577-19) as rare small nuclei can be observed in cell cultures independently of the presence of viral proteins. We now discuss this in Box 1, “Detecting and quantifying syncytia in vitro and in vivo” paragraph.
> Figure 3: Please add from which (published?) study/studies these picture are taken.
*** Authors’ response - We have now added more details regarding the source of these pictures in the legend of Figure 3. These pictures were prepared specifically for this review from material taken from past studies (a-b: Middleton et al. 2014, https://dx.doi.org/10.3201%2Feid2003.131159; c-f: Bossart et al. 2009, https://doi.org/10.1371/journal.ppat.1000642).
> Figure 4: What is the meaning of the included squares?
*** Authors’ response - Dark blue squares indicate that syncytia were observed but the exact proportion of individuals is unknown. This is indicated in the figure legend.
> Figure 5: Even though I was able to get the relevant information, I think the Map is difficult to read (space between the various cells is small). Could this be converted in a table?
*** Authors’ response - We have converted Figure 5 into a table (Table 2).
> Figure 6: Please adjust order in Figure itself with Figure Legend (e.g. start both times with Hooper et al., 1997b).
*** Authors’ response - We have re-ordered the panels on the Figure to match the legend (now Figure 5).
Minor points to consider:
> l. 2: In the Abstract authors state “other important viruses”. Important in what respect? Use a different word?
*** Authors’ response - We have changed “important” for “pathogenic” (line 2).
> l. 33: panencephalitis?
*** Authors’ response - We have corrected “panencephalitides” by “panencephalitis” (line 36).
> l. 73: word missing: … to understand the of syncytium…?
*** Authors’ response - We have corrected this sentence “to understand the contribution of syncytium formation” (line 76).
Figure legend to Figure 6: Spellings: Individuals received different doses and were (?) samples were collected from 2 to 17 days post infections (infection ?).
*** Authors’ response - We have corrected the typos pointed out (Figure 5 [formerly 6]).
Figure 7: Prevalence in %? 1 equals 100%?
*** Authors’ response - Indeed, prevalences were reported between 0 and 1 (corresponding to 0 to 100%). We now report prevalences in percentages in all the figures (Figures 4, 6 [formerly 7] and S3-7).
> l. 370: additional word? … 1994 HeV Australian HeV outbreak
*** Authors’ response - We have removed the first occurrence of “HeV” (lines 380).
> ll. 471 – 474. Can you please edit this sentence? This sentence is awfully long and nested.
*** Authors’ response - We have broken this sentence into two sentences now reading “This observation reinforces the idea that cell-cell fusion might represent, or be associated with a process representing, an evolutionary benefit. However, experimental evidence is lacking, and we cannot exclude that syncytium formation is a by-product of other viral functions (e.g., virus-cell fusion).” (lines 481-484).
> Funding: H.A.C should be H.C.A?
*** Authors’ response - Corrected (lines 594 and 595).
Reviewer 2 Report
Review of Viruses-1303101: Drivers and Distribution of Henipavirus-Induced Syncytia: What Do We Know?
Synopsis
The authors present an extensive review of syncytia formation induced by Henipavirus infection of cells in culture and in tissues of animal and human origin. The molecular prerequisites, including the interactions between the attachment glycoprotein, G, and their ephrin receptors, and the type I fusion protein, F, are outlined. Importantly, syncytia formation in vivo was reviewed in the context of different tissues, hosts, and viruses, which was very informative. Critically, the authors highlight the inconsistencies in data reporting and remark on how this can affect the interpretation of the importance of syncytia in pathogenesis. The review concludes with a discussion that focuses on the need for consistent reporting of the metrics used to quantify the properties of henipavirus induced syncytia in vitro and in vivo, plus, the need to investigate host factors that might regulate syncytia formation. Moreover, the authors propose whether syncytia formation can be used to predict viral fitness and pathogenicity.
General comments
This review is well written and provides excellent figures that collate the analyses of previous studies. The citations used for this review should be checked for relevance. The first two citations are not the most up to date with respect to cell-cell fusion in the herpesvirus field. For example, gE is not required for varicella-zoster virus (VZV) induced cell-cell fusion in vitro. I recommend replacing the first two reviews with 1,2. Another aspect of syncytia formation that was not touched upon was whether clinical isolates (animal or human) of viruses that induce syncytia are genetically homogenous. Recent work from the Szpara lab has demonstrated for herpes simplex viruses isolated from human patients are heterogenous populations with variations in syncytium forming properties 3. As syncytia are recognized across virus families it would be very informative to know whether there are heterogenous populations of henipaviruses within the host that have differentials in syncytia forming properties. If nothing is known about this phenomenon in the henipavirus or paramyxovirus arena it would be a useful addition to the ‘Conclusions’ section for future directions. The authors also raise the topic of cell factors in syncytia formation. This is an emerging topic in the herpesvirus field and for SARS-CoV-2 with cell factors that regulate cell-cell fusion starting to be identified 4-6. Although the authors do discuss cellular proteins such as ephrins and cathepsins, these are directly involved in the ability of the G and F proteins to induce membrane fusion. Have there been studies for henipaviruses or for members of the Paramyxoviridae that have implicated intracellular pathways in the regulation of syncytia formation? This would be another fascinating topic of brief discussion.
Specific comments and points to address
Lines 55-56, there is evidence that excessive syncytium formation is detrimental to herpesvirus pathogenesis 7-10. It will be helpful to make a statement and cite this literature.
Fig. 1a. The figure depicts that cell-cell fusion and syncytia formation predominantly occurs between infected and uninfected cells. Is this the case for henipaviruses? I acknowledge that this is the case when fusion is measured in vitro with transfection-based studies of viral glycoproteins and that there is evidence for virus spread via fusion. However, for VZV cell-cell fusion and syncytia formation is a rare event between infected and uninfected cells; typically, only infected cells will fuse 11,12. Can the authors please elaborate on syncytia formation between henipavirus infected and uninfected cells.
Figures 3, 4 and 5. It’s not clear whether these figures are the authors data or data collated from other studies. Please clarify this in the figure legends.
On Line 288 ‘pigs’ are referred to but in the previous sentence ‘Guinea pigs’ are referred to. Please clarify.
Line 347. How was viral antigen quantity measured?
Lines 353-355. Can the authors please elaborate on how this would be relevant in vivo.
Line 399. Typo “usable”.
Lines 483-487. See my comment for Lines 55-56.
References
1 Oliver, S. L., Zhou, M. & Arvin, A. M. Varicella-zoster virus: molecular controls of cell fusion-dependent pathogenesis. Biochem Soc Trans, doi:10.1042/BST20190511 (2020).
2 Connolly, S. A., Jardetzky, T. S. & Longnecker, R. The structural basis of herpesvirus entry. Nat Rev Microbiol, doi:10.1038/s41579-020-00448-w (2020).
3 Kuny, C. V., Bowen, C. D., Renner, D. W., Johnston, C. M. & Szpara, M. L. In vitro evolution of herpes simplex virus 1 (HSV-1) reveals selection for syncytia and other minor variants in cell culture. Virus Evol 6, veaa013, doi:10.1093/ve/veaa013 (2020).
4 Zhou, M., Kamarshi, V., Arvin, A. M. & Oliver, S. L. Calcineurin phosphatase activity regulates Varicella-Zoster Virus induced cell-cell fusion. PLoS Pathog 16, e1009022, doi:10.1371/journal.ppat.1009022 (2020).
5 Carmichael, J. C., Yokota, H., Craven, R. C., Schmitt, A. & Wills, J. W. The HSV-1 mechanisms of cell-to-cell spread and fusion are critically dependent on host PTP1B. PLoS Pathog 14, e1007054, doi:10.1371/journal.ppat.1007054 (2018).
6 Braga, L. et al. Drugs that inhibit TMEM16 proteins block SARS-CoV-2 spike-induced syncytia. Nature 594, 88-93, doi:10.1038/s41586-021-03491-6 (2021).
7 Oliver, S. L. et al. An immunoreceptor tyrosine-based inhibition motif in varicella-zoster virus glycoprotein B regulates cell fusion and skin pathogenesis. Proc Natl Acad Sci U S A 110, 1911-1916, doi:10.1073/pnas.1216985110 (2013).
8 Yang, E., Arvin, A. M. & Oliver, S. L. The cytoplasmic domain of varicella-zoster virus glycoprotein H regulates syncytia formation and skin pathogenesis. PLoS Pathog 10, e1004173, doi:10.1371/journal.ppat.1004173 (2014).
9 Engel, J. P., Boyer, E. P. & Goodman, J. L. Two novel single amino acid syncytial mutations in the carboxy terminus of glycoprotein B of herpes simplex virus type 1 confer a unique pathogenic phenotype. Virology 192, 112-120, doi:10.1006/viro.1993.1013 (1993).
10 Goodman, J. L. & Engel, J. P. Altered pathogenesis in herpes simplex virus type 1 infection due to a syncytial mutation mapping to the carboxy terminus of glycoprotein B. J Virol 65, 1770-1778, doi:10.1128/JVI.65.4.1770-1778.1991 (1991).
11 Reichelt, M., Brady, J. & Arvin, A. M. The replication cycle of varicella-zoster virus: analysis of the kinetics of viral protein expression, genome synthesis, and virion assembly at the single-cell level. J Virol 83, 3904-3918, doi:10.1128/JVI.02137-08 (2009).
12 Oliver, S. L., Yang, E. & Arvin, A. M. Dysregulated Glycoprotein B-Mediated Cell-Cell Fusion Disrupts Varicella-Zoster Virus and Host Gene Transcription during Infection. J Virol 91, doi:10.1128/JVI.01613-16 (2017).
Author Response
The authors present an extensive review of syncytia formation induced by Henipavirus infection of cells in culture and in tissues of animal and human origin. The molecular prerequisites, including the interactions between the attachment glycoprotein, G, and their ephrin receptors, and the type I fusion protein, F, are outlined. Importantly, syncytia formation in vivo was reviewed in the context of different tissues, hosts, and viruses, which was very informative. Critically, the authors highlight the inconsistencies in data reporting and remark on how this can affect the interpretation of the importance of syncytia in pathogenesis. The review concludes with a discussion that focuses on the need for consistent reporting of the metrics used to quantify the properties of henipavirus induced syncytia in vitro and in vivo, plus, the need to investigate host factors that might regulate syncytia formation. Moreover, the authors propose whether syncytia formation can be used to predict viral fitness and pathogenicity.
*** Authors’ response - We thank the reviewer for their positive evaluation of our manuscript and their suggestions, which considerably helped us improve our manuscript.
General comments
This review is well written and provides excellent figures that collate the analyses of previous studies. The citations used for this review should be checked for relevance. The first two citations are not the most up to date with respect to cell-cell fusion in the herpesvirus field. For example, gE is not required for varicella-zoster virus (VZV) induced cell-cell fusion in vitro. I recommend replacing the first two reviews with 1,2.
*** Authors’ response - We thank you for this input. We have updated the references as suggested (line 19 of the revised PDF).
Another aspect of syncytia formation that was not touched upon was whether clinical isolates (animal or human) of viruses that induce syncytia are genetically homogenous. Recent work from the Szpara lab has demonstrated for herpes simplex viruses isolated from human patients are heterogenous populations with variations in syncytium forming properties 3. As syncytia are recognized across virus families it would be very informative to know whether there are heterogenous populations of henipaviruses within the host that have differentials in syncytia forming properties. If nothing is known about this phenomenon in the henipavirus or paramyxovirus arena it would be a useful addition to the ‘Conclusions’ section for future directions.
*** Authors’ response - Unfortunately, very few henipavirus isolates are available and have been used in laboratory studies. In particular, there is not, to our knowledge, any study comparing in vitro or in vivo phenotypes of isolates within the same virus species, besides studies comparing the Bangladesh and Malaysia isolates of Nipah virus. In addition, studies comparing Nipah Bangladesh and Nipah Malaysia do so using live viruses, which makes it difficult to tease apart differences in cell-cell fusion phenotype versus differences in replication kinetics (resulting in differences in syncytium formation kinetics), as explained in Box 1. In any case, results vary across studies, potentially because of differences in protocols and/or cell lines. For instance, DeBuysscher et al. 2013 (https://dx.doi.org/10.1371%2Fjournal.pntd.0002024) reports that the Malaysian isolate exhibits faster replication and more intense cell-cell fusion than the Bangladesh isolate in baby hamster kidney (BHK-21) cells, while Griffin et al. 2019 (https://doi.org/10.1038/s41598-019-47549-y) reports no difference in Vero 6 cells. In vivo, the coverage and resolution of the data do not allow the detection of significant differences in cell-cell fusion between the two isolates (see Figure 4). We now discuss the need to better study cell-cell fusion across strains of a same virus, notably including recently sequenced bat and horse isolates (lines 504-508).
The authors also raise the topic of cell factors in syncytia formation. This is an emerging topic in the herpesvirus field and for SARS-CoV-2 with cell factors that regulate cell-cell fusion starting to be identified 4-6. Although the authors do discuss cellular proteins such as ephrins and cathepsins, these are directly involved in the ability of the G and F proteins to induce membrane fusion. Have there been studies for henipaviruses or for members of the Paramyxoviridae that have implicated intracellular pathways in the regulation of syncytia formation? This would be another fascinating topic of brief discussion.
*** Authors’ response - Studies conducted in henipaviruses and other paramyxoviruses suggest a key role of cytoskeleton (Johnston et al. 2019, https://doi.org/10.1128/mSystems.00194-19) as well as cell membrane composition (Countreras et al. 2021, https://doi.org/10.1128/JVI.02323-20) in cell-cell fusion, although comparative studies including several (henipa)viruses are lacking. We now mention these elements in the "Insights on syncytium formation from in vitro studies" section (lines 117-212).
Specific comments and points to address
Lines 55-56, there is evidence that excessive syncytium formation is detrimental to herpesvirus pathogenesis 7-10. It will be helpful to make a statement and cite this literature.
*** Authors’ response - We have added a sentence discussing the potential detrimental effect of excessive syncytium formation on pathogenesis, as well as the suggested references (lines 31-33).
Fig. 1a. The figure depicts that cell-cell fusion and syncytia formation predominantly occurs between infected and uninfected cells. Is this the case for henipaviruses? I acknowledge that this is the case when fusion is measured in vitro with transfection-based studies of viral glycoproteins and that there is evidence for virus spread via fusion. However, for VZV cell-cell fusion and syncytia formation is a rare event between infected and uninfected cells; typically, only infected cells will fuse 11,12. Can the authors please elaborate on syncytia formation between henipavirus infected and uninfected cells.
*** Authors’ response - In vitro studies suggest that henipavirus-induced syncytia can occur both between infected and non-infected cells (e.g., between effector and target cells in transfection studies, see Box 1), and between infected cells (notably syncytia fusing together, personal observations). We now explicitly state this in the legend of Figure 1.
Figures 3, 4 and 5. It’s not clear whether these figures are the authors data or data collated from other studies. Please clarify this in the figure legends.
*** Authors’ response - We have now added to the legends either the references (Figure 3) or that the data were extracted from the studies listed in Table S1 (Figure 4, S3 and S4), and references were directly added to Table 2 [formerly Figure 5].
On Line 288 ‘pigs’ are referred to but in the previous sentence ‘Guinea pigs’ are referred to. Please clarify.
*** Authors’ response - We now refer to pigs as “domestic pigs” to avoid potential confusion with Guinea pigs (lines 296, 376 and 448).
Line 347. How was viral antigen quantity measured?
*** Authors’ response - Viral antigens were quantified by immunohistochemistry in the cited study. We have added this information (line 355).
Lines 353-355. Can the authors please elaborate on how this would be relevant in vivo.
*** Authors’ response - Comparing virus replication dynamics in vitro in conditions in which cell-cell fusion is possible versus not possible would help understand whether syncytium formation has a positive effect on virus fitness. This would represent a first step into our investigation of the contribution of syncytium formation to pathogenesis, which depends on virus replication, spread and associated lesions. We now explicitly mention this lines 359-395.
Line 399. Typo “usable”.
*** Authors’ response - Corrected (line 409).
Lines 483-487. See my comment for Lines 55-56.
*** Authors’ response - We have added a sentence discussing the potential detrimental effect of excessive syncytium formation on pathogenesis (lines 31-33). We also point to the fact that the study of hyperfusigenic mutants (notably in vivo) can enrich our understanding of the mechanisms and consequences of syncytium formation (lines 493-495).
References
1 Oliver, S. L., Zhou, M. & Arvin, A. M. Varicella-zoster virus: molecular controls of cell fusion-dependent pathogenesis. Biochem Soc Trans, doi:10.1042/BST20190511 (2020).
2 Connolly, S. A., Jardetzky, T. S. & Longnecker, R. The structural basis of herpesvirus entry. Nat Rev Microbiol, doi:10.1038/s41579-020-00448-w (2020).
3 Kuny, C. V., Bowen, C. D., Renner, D. W., Johnston, C. M. & Szpara, M. L. In vitro evolution of herpes simplex virus 1 (HSV-1) reveals selection for syncytia and other minor variants in cell culture. Virus Evol 6, veaa013, doi:10.1093/ve/veaa013 (2020).
4 Zhou, M., Kamarshi, V., Arvin, A. M. & Oliver, S. L. Calcineurin phosphatase activity regulates Varicella-Zoster Virus induced cell-cell fusion. PLoS Pathog 16, e1009022, doi:10.1371/journal.ppat.1009022 (2020).
5 Carmichael, J. C., Yokota, H., Craven, R. C., Schmitt, A. & Wills, J. W. The HSV-1 mechanisms of cell-to-cell spread and fusion are critically dependent on host PTP1B. PLoS Pathog 14, e1007054, doi:10.1371/journal.ppat.1007054 (2018).
6 Braga, L. et al. Drugs that inhibit TMEM16 proteins block SARS-CoV-2 spike-induced syncytia. Nature 594, 88-93, doi:10.1038/s41586-021-03491-6 (2021).
7 Oliver, S. L. et al. An immunoreceptor tyrosine-based inhibition motif in varicella-zoster virus glycoprotein B regulates cell fusion and skin pathogenesis. Proc Natl Acad Sci U S A 110, 1911-1916, doi:10.1073/pnas.1216985110 (2013).
8 Yang, E., Arvin, A. M. & Oliver, S. L. The cytoplasmic domain of varicella-zoster virus glycoprotein H regulates syncytia formation and skin pathogenesis. PLoS Pathog 10, e1004173, doi:10.1371/journal.ppat.1004173 (2014).
9 Engel, J. P., Boyer, E. P. & Goodman, J. L. Two novel single amino acid syncytial mutations in the carboxy terminus of glycoprotein B of herpes simplex virus type 1 confer a unique pathogenic phenotype. Virology 192, 112-120, doi:10.1006/viro.1993.1013 (1993).
10 Goodman, J. L. & Engel, J. P. Altered pathogenesis in herpes simplex virus type 1 infection due to a syncytial mutation mapping to the carboxy terminus of glycoprotein B. J Virol 65, 1770-1778, doi:10.1128/JVI.65.4.1770-1778.1991 (1991).
11 Reichelt, M., Brady, J. & Arvin, A. M. The replication cycle of varicella-zoster virus: analysis of the kinetics of viral protein expression, genome synthesis, and virion assembly at the single-cell level. J Virol 83, 3904-3918, doi:10.1128/JVI.02137-08 (2009).
12 Oliver, S. L., Yang, E. & Arvin, A. M. Dysregulated Glycoprotein B-Mediated Cell-Cell Fusion Disrupts Varicella-Zoster Virus and Host Gene Transcription during Infection. J Virol 91, doi:10.1128/JVI.01613-16 (2017).
*** Authors’ response - We are very grateful for all the resources you shared in this review.